# Circulating miR-320a as a Predictive Biomarker for Left Ventricular Remodelling in STEMI Patients Undergoing Primary Percutaneous Coronary Intervention

**DOI:** 10.3390/jcm9041051

**Published:** 2020-04-08

**Authors:** Isabel Galeano-Otero, Raquel Del Toro, Agustín Guisado, Ignacio Díaz, Isabel Mayoral-González, Francisco Guerrero-Márquez, Encarnación Gutiérrez-Carretero, Sara Casquero-Domínguez, Luis Díaz-de la Llera, Gonzalo Barón-Esquivias, Manuel Jiménez-Navarro, Tarik Smani, Antonio Ordóñez-Fernández

**Affiliations:** 1Departamento de Fisiología Médica y Biofísica, Universidad de Sevilla, 41009 Sevilla, Spainrdeltoro-ibis@us.es (R.D.T.); 2Grupo de Fisiopatología Cardiovascular, Instituto de Biomedicina de Sevilla-IBiS, Universidad de Sevilla/HUVR/Junta de Andalucía/CSIC, Sevilla 41013, CIBERCV, 28029 Madrid, Spain; nnachoddcc@hotmail.com (I.D.); isabelmayoralgon@hotmail.com (I.M.-G.); gutierrez.encarnita@gmail.com (E.G.-C.); gonzalo.baron.sspa@juntadeandalucia.es (G.B.-E.); 3Servicio de Cardiología, Hospital Universitario Virgen del Rocío, 41013 Sevilla, Spain; aguiras@hotmail.com (A.G.); guerreromar24@gmail.com (F.G.-M.); saracasquero@gmail.com (S.C.-D.);; 4Hospital Universitario Virgen de la Victoria, Málaga 29010, CIBERCV, 28029 Madrid, Spain; mjimeneznavarro@gmail.com

**Keywords:** STEMI, PPCI, left ventricular adverse remodelling, circulating miRNAs

## Abstract

Restoration of epicardial coronary blood flow, achieved by early reperfusion with primary percutaneous coronary intervention (PPCI), is the guideline recommended to treat patients with ST-segment-elevation myocardial infarction (STEMI). However, despite successful blood restoration, increasing numbers of patients develop left ventricular adverse remodelling (LVAR) and heart failure. Therefore, reliable prognostic biomarkers for LVAR in STEMI are urgently needed. Our aim was to investigate the role of circulating microRNAs (miRNAs) and their association with LVAR in STEMI patients following the PPCI procedure. We analysed the expression of circulating miRNAs in blood samples of 56 patients collected at admission and after revascularization (at 3, 6, 12 and 24 h). The associations between miRNAs and left ventricular end diastolic volumes at 6 months were estimated to detect LVAR. miRNAs were also analysed in samples isolated from peripheral blood mononuclear cells (PBMCs) and human myocardium of failing hearts. Kinetic analysis of miRNAs showed a fast time-dependent increase in miR-133a, miR-133b, miR-193b, miR-499, and miR-320a in STEMI patients compared to controls. Moreover, the expression of miR-29a, miR-29b, miR-324, miR-208, miR-423, miR-522, and miR-545 was differentially expressed even before PPCI in STEMI. Furthermore, the increase in circulating miR-320a and the decrease in its expression in PBMCs were significantly associated with LVAR and correlated with the expression of miR-320a in human failing myocardium from ischaemic origin. In conclusion, we determined the time course expression of new circulating miRNAs in patients with STEMI treated with PPCI and we showed that miR-320a was positively associated with LVAR.

## 1. Introduction

ST-segment-elevation myocardial infarction (STEMI) is considered the most common and severe acute myocardial infarction (AMI) [1]. According to current guidelines, early reperfusion of coronary flow by primary percutaneous coronary intervention (PPCI) significantly mitigates cardiac cell death and cardiovascular events. PPCI has substantially reduced mortality following STEMI by decreasing the infarct size and the extent of left ventricular adverse remodelling (LVAR) [2,3]. However, increasing evidence has demonstrated that despite successful and prompt revascularization, ~25% of patients who survive STEMI develop heart failure (HF) as a consequence of LVAR [4,5,6,7]. STEMI treated with PPCI triggers a complex cascade of events associated with ischaemia and reperfusion (I/R), such as oxidative stress, neutrophil and platelet aggregation, and acute inflammatory responses [8] necessary to repair heart injuries. More specifically, peripheral blood mononuclear cells (PBMCs), mainly monocytes, participate in the acute response to heart damage. These cell populations egress from the bone marrow to the blood and infiltrate the myocardium hours after AMI [9,10]. Different studies have suggested that an exacerbated and prolonged inflammatory reaction causes irreparable damage which could be responsible for the progression of LVAR [11,12].

LVAR remains difficult to predict [13] even if there is general agreement regarding the value of troponin-I, creatine kinase (CK), and brain-derived natriuretic peptide (BNP) in post-acute coronary syndrome risk stratification [14]. However, none of the known cardiac biomarkers used today are considered reliable to predict the incidence of LVAR after PPCI in STEMI patients. Therefore, identifying new sensitive biomarkers may help to improve the early prognosis of patients with a high risk of post-infarction remodelling and dysfunction of the left ventricle after PPCI. Currently, there is great interest in the potential use of microRNAs (miRNAs) as promising novel biomarkers for the diagnosis and/or prognosis of different systemic diseases [15,16,17]. miRNAs are a group of small endogenous noncoding RNAs that can degrade mRNA transcripts directly, inhibit protein translation, and regulate protein expression at the post-transcriptional level [18]. miRNAs are present in blood and can be detected easily in serum or plasma due to their high stability (as reviewed previously [19]). A large number of studies have demonstrated that miRNAs are dysregulated in the blood of patients with cardiovascular diseases (CVD) compared to healthy controls. In fact, miRNAs have been suggested to be critical players in CVD since they participate in the genetic regulation of hundreds of key proteins involved in signalling pathways triggered by I/R [20,21,22]. Recent reports have described the expression patterns of circulating miRNAs in patients with myocardial infarction [23,24,25]. However, there are only a few studies describing miRNAs in STEMI patients after PPCI and their possible role as prognostic markers for the development of LVAR [26,27,28]. In this study, we examined the time course of the expression of miRNAs in STEMI patients after PPCI, and assessed the specific role of miR-320a as a prognostic biomarker of LVAR.

## 2. Materials and Methods

All procedures involving study subjects were performed according to the principles published by the declaration of Helsinki and its amendments or comparable ethical standards. The study was approved by the local Ethics Committee on Human Research at the University Hospital “Virgen del Rocio” of Seville and the University Hospital of “Virgen de la Victoria” of Málaga (approval no. 2013PI/096). Strengthening the reporting of observational studies in epidemiology (STROBE) guidelines were followed to report our findings (Appendix A).

### 2.1. Study Subjects and Blood Extraction

All study subjects voluntarily participated and signed an informed consent form. As depicted in Appendix A, a total of 56 participants who underwent coronary artery angiography at the above hospitals due to chest pain were selected. Patients were divided into 2 groups: (1) STEMI patients consisting of 42 patients who were diagnosed with STEMI for the first time and were treated with PPCI 1; 3 STEMI patients died before finishing this study, and (2) Control group consisting of 14 non-STEMI patients whose coronary angiogram showed no coronary lesion. The inclusion criteria were as follows: patients under 75 years old, diagnosed with AMI, presenting symptoms 2 to 6 h prior to angioplasty and exhibiting occlusion of the left anterior descending (LAD) artery with epicardial blood flow (“Thrombolysis in Myocardial Infarction” (TIMI) flow grade) of 0 in the initial angiogram. The exclusion criteria were as follows: patients with a previous history of ischaemic heart disease, a glomerular filtration rate less than 30 mL/min, TIMI flow > 0–1 at the time of angiography. Patients received standard pharmacological therapy as per current clinical guidelines. Information relative to clinical, demographical, haemodynamic, angiographic and electrocardiographic findings was prospectively registered in all cases upon admission. In some experiments, blood samples from healthy volunteers without known disease were used as a healthy control to compare the expression of miRNAs.

First, blood samples were extracted after catheter insertion in the radial artery before initiation of the PPCI procedure. These samples represented the pre-reperfusion time point 0 h. Additional blood samples were collected at 3, 6, 12 and/or 24 h after culprit vessel opening. Additional blood samples were collected 1 month after the ischaemic event. Echocardiography was performed to determine ventricular ejection fraction and systolic and diastolic left ventricular diameters and volumes before patient discharge and 6 months after PPCI to determine the possible development of LVAR. LVAR was defined as an increase of at least 20% in left ventricular end-diastolic volume (LVEDV) with left ventricular ejection fraction less than 50%, as defined according to current clinical indications [29].

Serum was obtained from whole blood samples collected without antiserum. PBMCs were isolated from blood using ethylenediaminetetraacetic acid (EDTA)-coated tubes. The lymphocyte separation media (Lymphosep; Biowest, Riverside, MO, USA) was used to isolate PBMCs from blood following the manufacturer’s instructions.

### 2.2. Human Samples

Myocardial biopsies were obtained from the atrium of ischaemic patients with heart failure (HF) during cardiac surgery. The 7 HF patients (5 males and 2 females) had a median age of 62 years, and their ejection fraction (EF) before surgery was 52% ± 12%. Fully informed written consent was obtained from the families of all donors.

### 2.3. RNA Isolation and miRNA Analysis

Blood samples were collected before the start of the PPCI procedure and at multiple time points following PPCI. We used the miRNeasy Serum/Plasma kit (QIAGEN, Hilden, Germany) to extract small RNAs from serum. Briefly, 200 μL of serum was mixed with 700 μL of QIAzol Lysis Reagent included in the kit, and 10 pmol of Arabidopsis miRNA (ath-miR-159a) was added as a spike-in control. Then, we followed the manufacturer’s instructions to obtain the fraction of eluted miRNAs. To obtain small RNAs from PBMCs, we used the mirVana miRNA Isolation Kit (Thermo Fisher Scientific Inc., Waltham, MA, USA). The eluted miRNAs were quantified using a fluorometer (Qubit 4 Thermo Fisher Scientific Inc., Waltham, MA, USA) through a Qubit miRNA assay (Thermo Fisher Scientific Inc., Waltham, MA, USA). We used TaqMan array miRNA card pool A (Applied Biosystems Thermo Fisher Scientific Inc., Waltham, MA, USA) to examine miRNA expression in 9 STEMI and 3 control patients. Once we had selected miRNA candidates, custom TaqMan miRNA arrays (Applied Biosystems, Thermo Fisher Scientific Inc., Waltham, MA, USA) were designed to amplify miRNAs. To validate miRNA expression, we used RT-qPCR using the TaqMan Advanced miRNA cDNA Synthesis Kit, TaqMan Advanced miRNA Assay, and TaqMan Fast Advanced Master Mix technology. To determine miRNA expression in PBMCs, we used the miScript II RT Kit (QIAGEN, Hilden, Germany) and iTaq Universal SYBR Green Supermix (Bio-Rad, Hercules, CA, USA ). RT-qPCR was performed using an Applied Biosystems Viia7 7900HT thermocycler. Relative quantification analyses were performed using the software SDS2.2 and Expression Suite Software V1.0.3 (Applied Biosystems, Thermo Fisher Scientific Inc., Waltham, MA, USA), online software ThermoFisher Cloud and QuantStudio Real Time PCR Software (Thermo Fisher Scientific Inc., Waltham, MA, USA) and Excel. Fold changes in miRNA expression were calculated using the comparative cycle threshold CT (ΔΔCT) method. The values are expressed as the logarithm of the fold change.

### 2.4. In Silico miRNA Studies

We used the miRDB (miRDB v7.2, http://mirdb.org, Washington University St. Louis, MO, USA) and TargetScan (www.targetscan.org, Cambridge, MA, USA) databases to analyse miRNA targets. We found common targets between the databases through a Venn diagram. To identify microRNA target gene pathways, we used an online platform from the Gene Ontology (GO) browser PANTHER (Protein Analysis THrough Evolutionary Relationships 14.1 version, http://pantherdb.org/genelistanalysis.do, University of Southern California, Los Angeles, CA, USA). 

### 2.5. Statistical Analysis

Data were analysed using SPSS (SPSS Inc. version 25.0 IBM, Armonk, NY, USA) and with GraphPad (GraphPad Software Inc., San Diego, CA, USA). The results are presented as the mean and standard error of the mean (SEM). The outliers were removed based on the results of QuickCalcs, an online tool of GraphPad. The Shapiro–Wilk test was used for normality. For normally distributed variables, we used an ordinary one-way ANOVA, and we performed multiple comparisons using T-test without correction (Fisher’s LSD test). For non-normal distribution, we used the non-parametric Kruskal–Wallis test with multiple comparisons corrected by Dunn’s test. Multivariate logistic regression analysis was performed to estimate the independent relationship between variables with significant differences (*p* < 0.1) and the appearance of LVAR. LVAR was taken as the binary dependent variable, miR-320a expression as the independent, and CK, sex and age as the covariates for adjustment. Linear regression analysis was conducted using the percentage of change in LVEDV as dependent variable. The LVAR predictive value of different parameters (miR-320a expression, CK, and age) was also evaluated using the receiver operating characteristic (ROC) curve.

## 3. Results

### 3.1. Analysis of the Clinical Data of the Subjects

Clinical information about the study subjects is shown in Table 1. A total of 42 patients with STEMI who underwent PPCI and 14 controls were included in this study. Table 1 indicates that 88% of the STEMI patients were male, and there were no significant differences between STEMI and control patients (*p* > 0.05) in terms of age, risk factors and LVEDV index at the time of admission. In contrast, we observed significant differences in cardiac markers, creatine kinase and troponin T between both groups at the onset of PPCI.

### 3.2. miRNA Expression Profiles in STEMI

Samples from control and STEMI patients with TIMI 0 flow were used to detect the profiles of miRNAs secreted in the serum 3 to 6 h after PPCI by a RT-qPCR-based array. Figure 1A shows volcano plot analysis indicating significant alterations in the expression of miRNAs. The hierarchical clustering analysis indicated that 25 miRNAs were differentially expressed despite the variability between STEMI patient samples compared to the control group (Figure 1B). Of these, 96 miRNAs were upregulated and 138 were downregulated. Based on this finding, we selected 5 miRNAs (miR-193b, miR-320a, miR-339-5p, miR-522, and miR-545) to examine their expression in serum taken from 8 patients with STEMI at different time points: before PPCI (0 h) and 3, 12, and 24 h after revascularization. Samples from 8 non-STEMI patients were used as controls.

We also examined the levels of other miRNAs (miR-1, miR-21, miR-29a, miR-29b, miR-125, miR-133a, miR-133b, miR-208, miR-324, miR-423-5p and miR-499), selected based on a literature search focusing on their relevance in different CVD [26,30,31,32,33,34] using the search term “miRNA” in combination with one of the following key words: “STEMI”, “ischaemia and reperfusion”, “heart infarction”, and “heart failure”. Figure 2A–E shows a significant increase in the expression of miR-133a, miR-133b, miR-193b, miR-499, and miR-320a in STEMI patients at different time points compared to the control group. The levels of these miRNAs mainly reached a maximum increase between 3 and 12 h and returned to their basal level 24 h after PPCI. In the case of miR-423, miR-29a, miR-339-5p, and miR-324, their expression increased and remained significantly higher 24 h after PPCI (Figure 2F–I). In contrast, Figure 3A–D shows that the expression of miR-29b, miR-208, miR-522, and miR-545 was significantly downregulated soon after PPCI and continued to be downregulated at all examined time points in STEMI patients. Moreover, Figure 3E,F show that miR-1 levels were downregulated only at the time point before PPCI, while miR-21 seemed to not be sensitive to PPCI.

### 3.3. miRNA Expression Pattern in STEMI with LVAR

To further characterize the relevance of the observed changes in the expression of miRNAs, we examined whether their levels were different in patients who developed LVAR 6 months after PPCI compared to those who did not present any adverse events after PPCI interventions. Table 2 shows that of 39 patients, 14 (22%) developed LVAR since their LVEDV index values increased significantly 6 months after PPCI compared to the values at admission. We also observed a significant difference in the level of creatine kinase, but not troponin T, at the time of admission of patients with or without LVAR. Moreover, there were no significant differences between the groups in terms of age, gender, risk factors, or pro-BNP 6 months after PPCI (*p* > 0.05).

Next, based on our previous results, we compared the expression of miR-320a, miR-193b, miR-324, miR-339-5p, miR-519a, miR-522, and miR-545 in STEMI patients (*n* = 8) with or without LVAR (*n* = 8). Figure 4A shows that the levels of miR-320a were significantly increased at 3 and 12 h after PPCI in LVAR patients compared to patients without remodelling. Moreover, no differences in the levels of the other miRNAs were observed between the groups (Figure 4B–F). To confirm that the expression levels of miR-320a were associated with the appearance of LVAR, we examined miR-320a expression in a larger cohort of patients (*n* = 39). In this case, we considered 6 h as the optimal time point to test the miRNA maximum levels after PPCI, and we also evaluated its expression 1 month after revascularization. Figure 5A confirms a transient rise of miR-320a in STEMI patients, reaching a maximum increase 6 h after PPCI, while its expression decreased significantly 1 month after patient discharge. Furthermore, the comparison of miR-320a levels between patients with or without LVAR (Figure 5B) shows that miR-320a was still significantly higher 1 month after discharge in the non-remodelling group but not in LVAR patients. This fact suggests that changes in the levels of circulating miR-320a can be associated with the appearance of LVAR. In another set of experiments, we determined the expression of miR-320a in healthy volunteers and compared it with the expression in the control group. Appendix A shows that the levels of miR-320a were not significantly different between the two groups, and the expression of miR-320a in STEMI patients with or without LVAR compared to its levels in healthy controls (Appendix A) was similar to those observed when compared to the clinical control (Figure 5B).

### 3.4. Correlation of Serum Levels of miR-320a with LVAR

Linear regression and ROC analysis were performed to further confirm that changes in circulating levels of miR-320a associates with LVAR. Figure 6A shows that the levels of miR-320a at 1 month (r = 0.651, *p* = 0.03) correlated inversely and significantly with the percentage of changes in LVEDV, meanwhile CK (r = 0.462, *p* = 0.053) and age (r = 0.085, *p* = 0.738) showed worse association with LVEDV changes. Moreover, ROC curve analysis in Figure 6B shows that the area under the curve (AUC) of miR-320a expression was 0.889 (95% CI: 0.74–1.00; *p* value = 0.004), while the AUC of CK was 0.722 (95%CI: 0.463–0.981; *p* value = 0.102), and of age was 0.530 (95%CI: 0.251–0.809; *p* value = 0.819) (Appendix A). ROC analysis also indicated a sensitivity of 70% and a specificity of 88.89% for patients with the levels of miR-320a < −0.306. Furthermore, as shown in Table 3, multivariate analysis identified miR-320a as an independent predictor of LVAR (*p* < 0.045).

### 3.5. Expression of miR-320a in PBMCs Isolated from STEMI and in the Myocardium of Ischaemic Heart Failure Patients

Since AMI involves a complex cascade of events that activate acute inflammatory responses [8], we examined the possible source of the fast increase in circulating miRNAs and focused on PBMCs. Figure 7A shows that miR-320a was slightly upregulated in the PBMCs of patients suffering from STEMI even before revascularization (time point 0 h), compared to the control patients. Importantly, and in contrast to what we observed in serum samples (Figure 5A), the levels of miR-320a decreased significantly in PBMCs at 6 h after PPCI. Moreover, as shown in Figure 7B, the comparison of miR-320a in patients with or without LVAR indicated that miR-320a decreases significantly at 6 h after PPCI only in patients with LVAR, while its expression did not change in non-LVAR patients. Appendix A also confirmed that the expression of miR-320a in PBMCs from the control group was not different than that of healthy volunteers and shows similar changes in STEMI patients with or without LVAR (Appendix A). Moreover, the multivariate analysis indicates that expression levels of miR-320a in PBMCs showed a trend association, although not significant, with LVAR (OR, 0.052; 95% CI, 0.02–1.281; *P* = 0.071 at 6 h, and OR, 0.206; 95% CI, 0.037–1.140; *P* = 0.070 at 1 month post PPCI).

Finally, we examined the expression of miR-320a and other miRNAs in patients with HF of ischaemic origin. Figure 8A shows significant expression of miR-320a in the myocardial tissue in the atrium (*n* = 7). As illustrated in Figure 8B, compared to the expression of other miRNAs, the expression of miR-320a was significantly higher than that of miR-324-5p, miR-324-3p, miR-339-5p, and miR-423 but not that of miR-29a or miR-499-5p. In contrast, miR-320a levels were lower than miR-133 levels.

### 3.6. Analysis of miR-320a Target Genes

Having confirmed the association of miR-320a with the appearance of LVAR, we next searched relevant miR-320a target genes in the context of AMI. To this end, we performed an in silico analysis using miRDB and TargetScan based on the miR-320a sequence, and compared the common genes identified in both databases. Using PANTHER software (Protein Analysis THrough Evolutionary Relationships 14.1 version, http://pantherdb.org/genelistanalysis.do, University of Southern California, Los Angeles, CA, USA), we generated a graphic showing that 1060 genes and 520 pathways were mainly implicated in the processes regulating adverse remodelling (Figure 9). As highlighted, miR-320a was predicted to target 13 genes associated with apoptosis, 25 with the fibroblast growth factor (FGF) and tumour growth factor (TGF)-beta signalling pathways, 18 with inflammation, and 3 with oxidative stress. Altogether, these results suggest that miR-320a could have an important role in ventricular adverse remodelling through the regulation of different signalling pathways implicated in AMI.

## 4. Discussion

AMI is a complex syndrome that has become a major public health problem due to its high morbidity and mortality [35,36]. Therefore, there is an increasing need to identify new diagnostic and therapeutic biomarkers for patients with AMI, even those with successful revascularization. In fact, despite the reestablishment of coronary artery perfusion, there is still an increasing incidence of LVAR in STEMI patients who underwent successful PPCI. In agreement with the literature [4,7], almost 22% of our STEMI patients developed LVAR as early as 6 months after PPCI. Classical cardiac markers, such as troponin-T and creatine kinase, were increased in all patients suffering from STEMI at admission, confirming the severity of the infarcts. We also found that creatine kinase, but not troponin T, was significantly higher in patients who developed LVAR compared with those without LVAR, consistent with a larger infarct size, which is in agreement with other studies [37,38].

The main findings of this study are as follows: (i) miRNAs are quickly released into the circulation in different patterns following successful coronary artery revascularization in STEMI patients; (ii) a rapid increase in circulating miR-320a was observed in STEMI patients; (iii) circulating miR-320a decrease 1 month post PPCI correlates with the appearance of LVAR; (iv) the levels of miR-320a in PBMCs correlates inversely with expression in blood; and (v) higher expression of miR-320a is detected in the heart tissue of patients with HF of ischaemic origin.

In recent years, circulating miRNAs in patients with myocardial infarction have been extensively studied [23,25,39]. For instance, miR-150 [40], miR-208b, miR-34a [32], miR-328 and miR-134 [41] have been suggested as biomarkers of LVAR after AMI. However, little is known regarding miRNAs in AMI patients after revascularization. The patients selected for this study were a homogenous group of patients who were admitted to the hospital with 2 to 6 h of chest pain and no previous history of ischaemic heart disease, showing an initial TIMI flow of 0 in the LAD coronary artery after an angiogram. The inclusion criteria were rigorous to ensure consistent results related to the impact of reperfusion on miRNA expression. In contrast, other recent studies examined miRNAs in patients with a TIMI flow of 2 or 3, which may not reflect complete ischaemia [27,42]. Our study describes the expression profile of circulating miRNAs in STEMI patients examined at early time points after successful revascularization. This method provided an important assessment of how fast the levels of circulating miRNAs changed in the blood after STEMI, consistent with a recent study that nicely demonstrated monophasic and biphasic kinetic patterns of circulating miRNAs following myocardial reperfusion in STEMI patients [42]. It is well known that miRNA expression is tissue-specific and responds rapidly to changes in the body, which suggests their usability as biomarkers for the diagnosis and prognosis of diseases [23]. Here, we show that the levels of some miRNAs were significantly lower or higher at admission, indicating that these changes were triggered by ischaemia, while other miRNAs increased in a time-dependent manner after the PPCI procedure, reaching maximum levels between 3 and 12 h, which is in agreement with a recent study [42].

Recent studies on the use of PPCI described that miRNAs might also predict worse outcomes of patients who underwent PPCI. For example, miR-1 and miR-133b levels increased within 3 h of PPCI, and these miRNAs are positively associated with microvasculature obstruction and worse left ventricular functional recovery [42]. miR-1254 levels at admission predict volume changes of the left ventricle and post-STEMI left ventricular remodelling after PPCI [27]. In contrast, lower levels of miR-30e at admission are associated with no reflow in STEMI patients undergoing PPCI [43]. Here, in this study, we demonstrate for the first time that only miR-320a, which peaks at 6 h after PPCI, predicts the change in LVEDV determined at 6 months after patient discharge. According to multivariate and ROC analysis, miR-320a was independently associated with changes in LVEDV, exhibiting quite accurate sensitivity and specificity. Moreover, our data showed that miR-320a, but not CK nor age, can reliably predict the occurrence of LVAR. Nevertheless, these data deserve to be confirmed in a larger cohort of patients.

To identify the putative source of this fast increase in the level of miR-320a, we examined its expression in PBMCs since these cells have been implicated in the early inflammatory process (as reviewed elsewhere [10]). Our data indicated that miR-320a was significantly increased in PBMCs of the patients suffering from STEMI at admission before revascularization compared to the controls. Importantly, 6 h after PPCI, miR-320a levels decreased in PBMCs, which was inversely correlated with their levels in the serum of the same patients. Furthermore, the comparison of miR-320a expression in patients with or without LVAR indicated that its lower expression in PBMCs was associated with LVAR. Interestingly, we show for the first time that the human failing myocardium from ischaemic origin significantly expresses miR-320a. In these patients, we observed that the expression of miR-320a was generally higher than that of other examined miRNAs associated with heart diseases. A previous report indicated that miR-320 is likely overexpressed in human left ventricular samples of the patients with ischaemic cardiomyopathy and aortic stenosis [44], which supports our findings. Of note, miR-320a remained at a lower level both in serum and PBMCs one month after PPCI only in patients who will develop LVAR, probably because miR-320a is continuously delivered to the myocardium. Altogether, these data suggest that PBMCs from patients who will suffer adverse remodelling may express miR-320a in the blood and presumably in the myocardium, predicting a worse prognosis. This finding needs deeper investigation to demonstrate the precise mechanism by which PBMCs may release miR-320a into the bloodstream and to cardiac tissue. Emerging studies have indicated that blood contains large numbers of extracellular vesicles that transport miRNAs to different tissues in organisms [45], such as exosomes, originated from platelets, erythrocytes, leucocytes, and vascular cells, particularly the endothelium [46,47,48]. Recently, miR-320a has been detected in myocardial microvascular endothelial cells from a Goto–Kakizaki (GK) diabetic rat model [49]. Previous studies regarding the function of miR-320a in the myocardium have demonstrated its abnormal expression in animal heart hypertrophy and human heart disease [50,51,52]. miR-320 is presumably upregulated in the heart by hyperglycaemia, which acts through CD36 (fatty acid translocase) transcription [53]. Other studies determined that miR-320 overexpression is linked to cell death and apoptosis through heat-shock protein 20 [50] and insulin growth factor-1 [54], or by targeting AKIP1 and inducing the mitochondrial apoptotic pathway [55], among other mechanisms. Our results from in silico and computational miRNA target prediction algorithms revealed that miR-320a can regulate the expression of genes implicated in cardiac signalling pathways activated under ischaemia. For instance, the miR-320a predicted target genes were associated with the FGF and TGF signalling pathways, whose roles are widely known in cardiac fibrosis and wound healing [56,57]. Other target genes were associated with apoptosis and oxidative stress critical events that occur in ischaemia and reperfusion [58,59], and yet others were linked to inflammation mediated by chemokines that coincide the impact of AMI and revascularization [12]. Overall, it is clear that miR-320a upregulation may cause deleterious cellular events, and its downregulation may be involved in protection against cardiac remodelling, which will promote the improvement of heart function. Nevertheless, miR-320a-activated molecular mechanisms in the context of early revascularization are still not completely understood and deserve further investigation.

## 5. Conclusions

This study shows detailed miRNA release kinetics in the initial hours following PPCI. This analysis provides evidence for miR-320a as a potential prognostic biomarker for LVAR in patients who have undergone PPCI, which is promising for the preventive treatment of clinical heart failure. Further studies are needed to establish the pathophysiological and clinical significance of increased circulating miRNA-320a levels and to investigate the therapeutic efficacy of targeting miRNA-320a.

### Study Limitations and Clinical Perspectives

The present study aimed to validate the proof-of-concept that circulating miRNAs can predict the outcome of STEMI patients undergoing PPCI.

In evaluating the results of this study, one must take several limitations into account:

The small sample size used in this study, due to the difficulty of recruiting patients with homogeneous criteria. Patients are only from two hospitals, therefore, a large-scale and multicentre study would be welcomed to confirm the role of miRNA-320a as a potential biomarker for STEMI patients with LVAR.

The results are limited only to patients with early revascularized STEMI with TIMI flow 0.

Our findings are essentially limited to a change in the volume of the left ventricle after revascularized STEMI up to six months, and remodelling may increase afterwards.

The specific roles of miRNAs have not been evaluated in terms of the ability to regulate gene expression in human cardiac myocytes.

It is possible that other miRNAs may also have prognostic value in this context, since RNA sequencing or microarray approaches are continuously improving in terms of sensitivity and specificity.

## Figures and Tables

**Figure 1 jcm-09-01051-f001:**
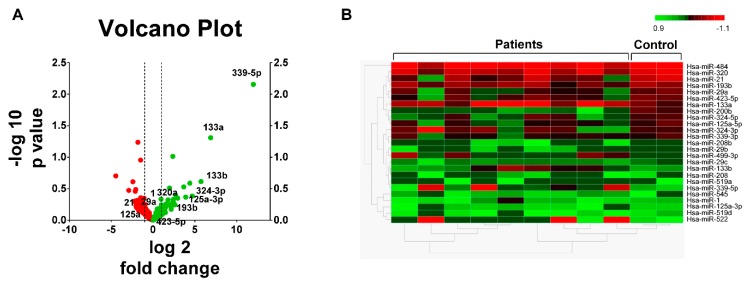
miRNAs differentially expressed in STEMI patients 3 h post PPCI. (**A**) Volcano plot showing miRNAs differentially expressed in STEMI patients (*n* = 9) relative to the control (*n* = 3); the x axis shows log_2_ (fold change), and the y-axis shows –log_10_ (*p* value). (**B**) Fragment of hierarchical clustered sample-centric heat-map analysis of the ΔCt value of differentially expressed miRNAs in STEMI patients 3 h post PPCI compared to the control. Scale bar: downregulated (red) and upregulated (green). Distance was measured by Pearson’s correlation.

**Figure 2 jcm-09-01051-f002:**
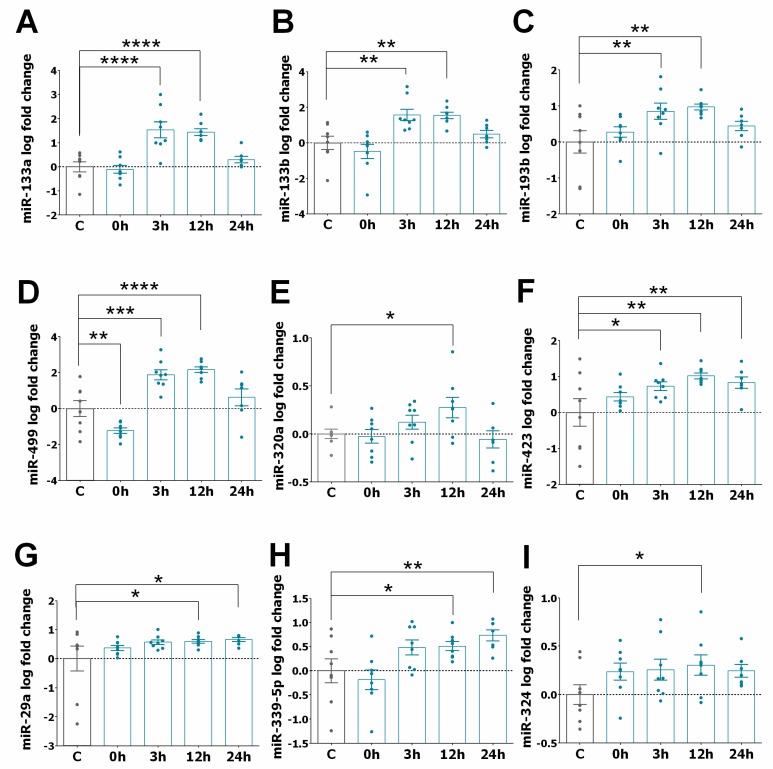
Upregulation of circulating miRNAs in the serum of post-STEMI patients. (**A–I**) Graphs showing the kinetics of miRNAs in serum samples of STEMI patients (*n* = 8) before PPCI (0 h) and 3, 12, and 24 h after the PPCI procedure. Bar graphs show changes in the expression of miR-133a (**A**), miR-133b (**B**), miR-193b (**C**), miR-499 (**D**), miR-320a (**E**), miR-423 (**F**), miR-29a (**G**), miR-339-5p (**H**) and miR-324 (**I**) in patients after PPCI compared to control patients (*n* = 8). Values represent the fold changes (in logarithmic scale) for each miRNA relative to controls. Data are presented as the means ± SEM. Significance is indicated by (*) for *p* < 0.05, (**) for *p* < 0.01, (***) for *p* < 0.001), and (****) for *p* < 0.0001. Ordinary one-way ANOVA with multiples comparisons using T-test without correction (Fisher’s LSD test) was performed.

**Figure 3 jcm-09-01051-f003:**
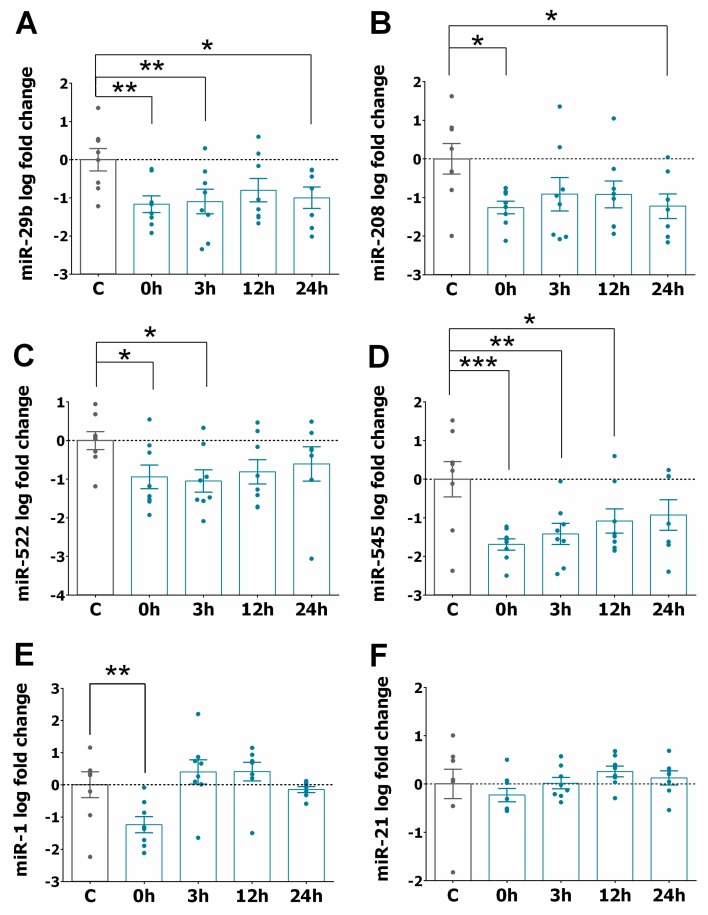
Downregulation of circulating miRNA levels in the serum of STEMI patients. (**A–F**) Graphs represent miRNA expression in the serum of STEMI patients (*n* = 8) compared to the control (*n* = 8) before PPCI (0 h) and 3, 12, and 24 h. Bar graphs show the expression of miR-29b (**A**), miR-208 (**B**), miR-522 (**C**), miR-545 (**D**), miR-1 (**E**) and miR-21 (**F**) in patients compared to control patients. Values are presented as the means ± SEM and represent the fold change in logarithmic scale for each miRNA relative to controls. Significance is indicated by (*) for *p* < 0.05, (**) for *p* < 0.01, and (***) for *p* < 0.001). Ordinary one-way ANOVA with multiples comparisons using T-test without correction (Fisher’s LSD test) was performed.

**Figure 4 jcm-09-01051-f004:**
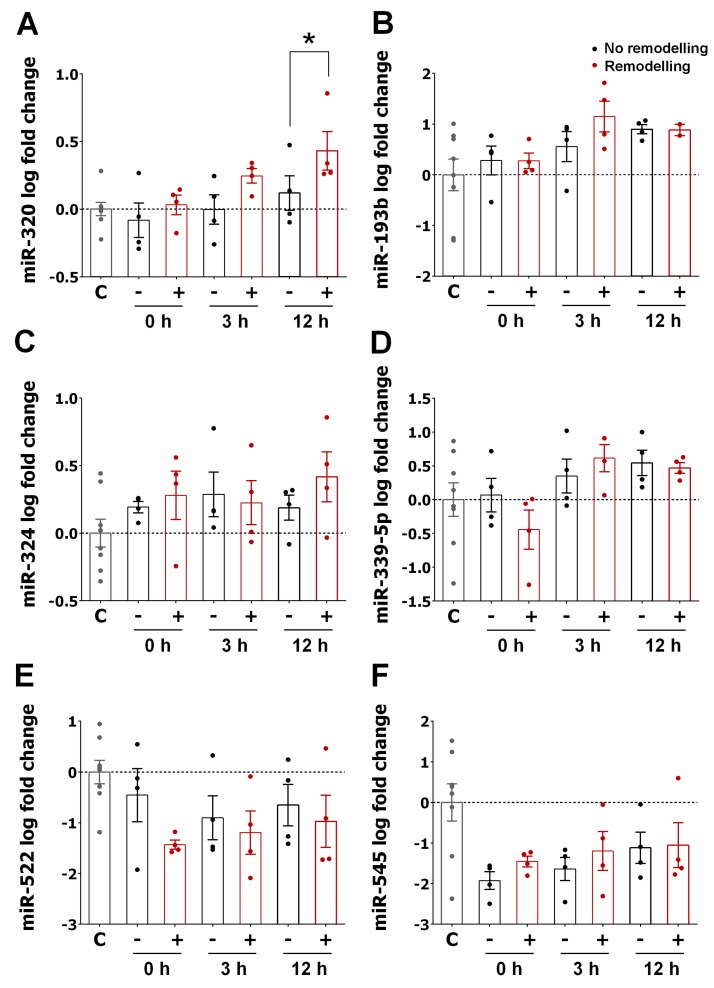
Comparison of the expression of miRNAs in patients with or without left ventricular adverse remodelling. (**A–F**) Bar graphs show the expression of miR-320a (**A**), miR-193b (**B**), miR-324 (**C**), miR-339-5p (**D**), miR-522 (**E**), and miR-545 (**F**) in serum samples of STEMI patients who developed left ventricular adverse remodelling (red bars, *n* = 4) or not (black bars, *n* = 4) before PPCI (0 h) and at 3, 12, and 24 h after the procedure. The values in the graphs represent the fold change in logarithmic scale for each miRNA relative to controls. Values are presented as the means ± SEM and represent the fold change in logarithmic scale for each miRNA relative to controls (*n* = 8). Significance is indicated by (*) for *p* < 0.05. Ordinary one-way ANOVA with multiples comparisons using T-test without correction (Fisher’s LSD test) was performed.

**Figure 5 jcm-09-01051-f005:**
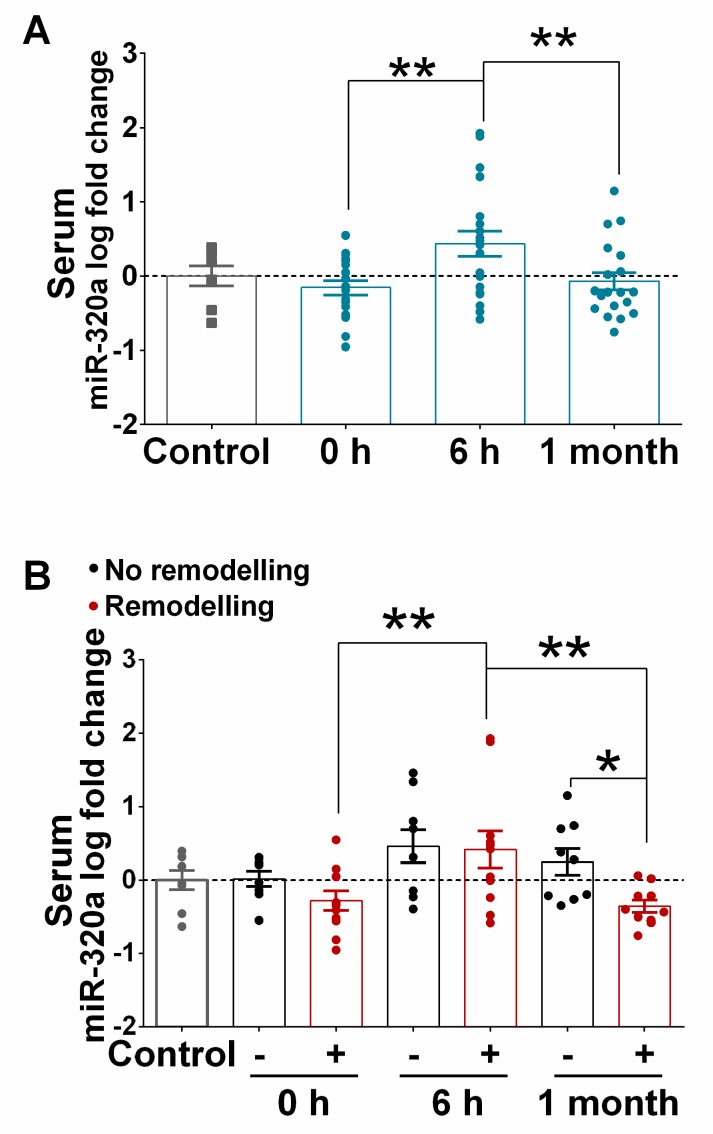
Comparison of the expression of miR-320a in the serum of patients with or without left ventricular adverse remodelling. (**A**) Bar graph shows miR-320a expression in serum samples of STEMI patients (*n* = 20) at 0 h, 6 h and 1 month post PPCI relative to the control (*n* = 8). (**B**) Bar graph shows the levels of miR-320a at 0 h, 6 h and 1 month after PPCI in STEMI patients who developed LVAR (red bar, *n* = 11) or not (black bar, *n* = 9). Values in the graphs represent the fold change in logarithmic scale for each miRNA relative to controls. Data are the means ± SEM. Significance is indicated by (*) for *p* < 0.05 and (**) for *p* < 0.01. Ordinary one-way ANOVA with multiples comparisons using T-test without correction (Fisher’s LSD test) was performed.

**Figure 6 jcm-09-01051-f006:**
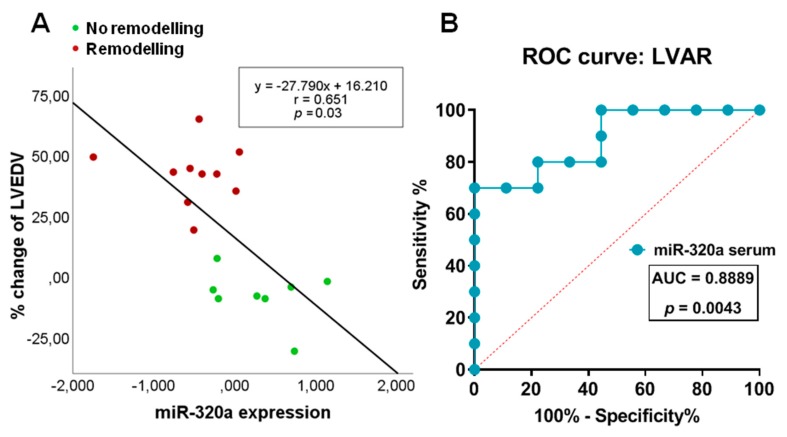
Correlation of serum levels of miR-320a with left ventricular adverse remodelling (LVAR) and receiver-operating characteristics (ROC) analysis. (**A**) Linear regression analysis using the percentage of change in left ventricular end-diastolic volume (LVEDV) as dependent variable and miR-320a expression 1 month post-PPCI as independent. (**B**) The area under the curve (AUC, values given on the graphs) analysis of ROC indicating sensitivity and specificity of miR-320a in predicting LVAR.

**Figure 7 jcm-09-01051-f007:**
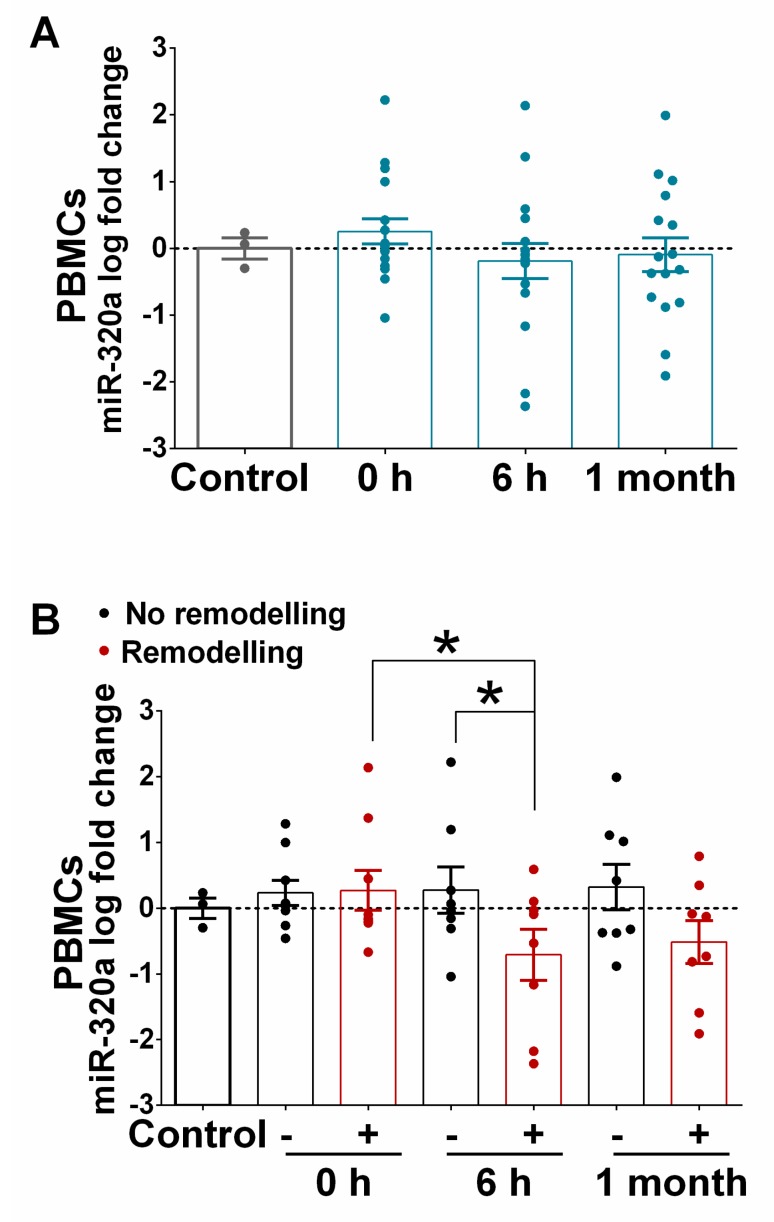
Expression of miR-320a in peripheral blood mononuclear cells (PBMCs) of patients with or without left ventricular adverse remodelling. (**A**) Bar graph shows miR-320a expression in PBMCs of STEMI patients (*n* = 20) at 0 h, 6 h and 1 month post PPCI relative to the control (*n* = 8). (**B**) Bar graph shows the levels of miR-320a in PBMCs at 0 h, 6 h and 1 month after PPCI in STEMI patients who developed LVAR (red bar, *n* = 11) or not (black bar, *n* = 9). Values in the graphs represent the fold change in logarithmic scale for each miRNA relative to the controls. Data are the means ± SEM. Significance is indicated by (*) for *p* < 0.05. Kruskal–Wallis test with multiple comparisons corrected by Dunn’s test was performed.

**Figure 8 jcm-09-01051-f008:**
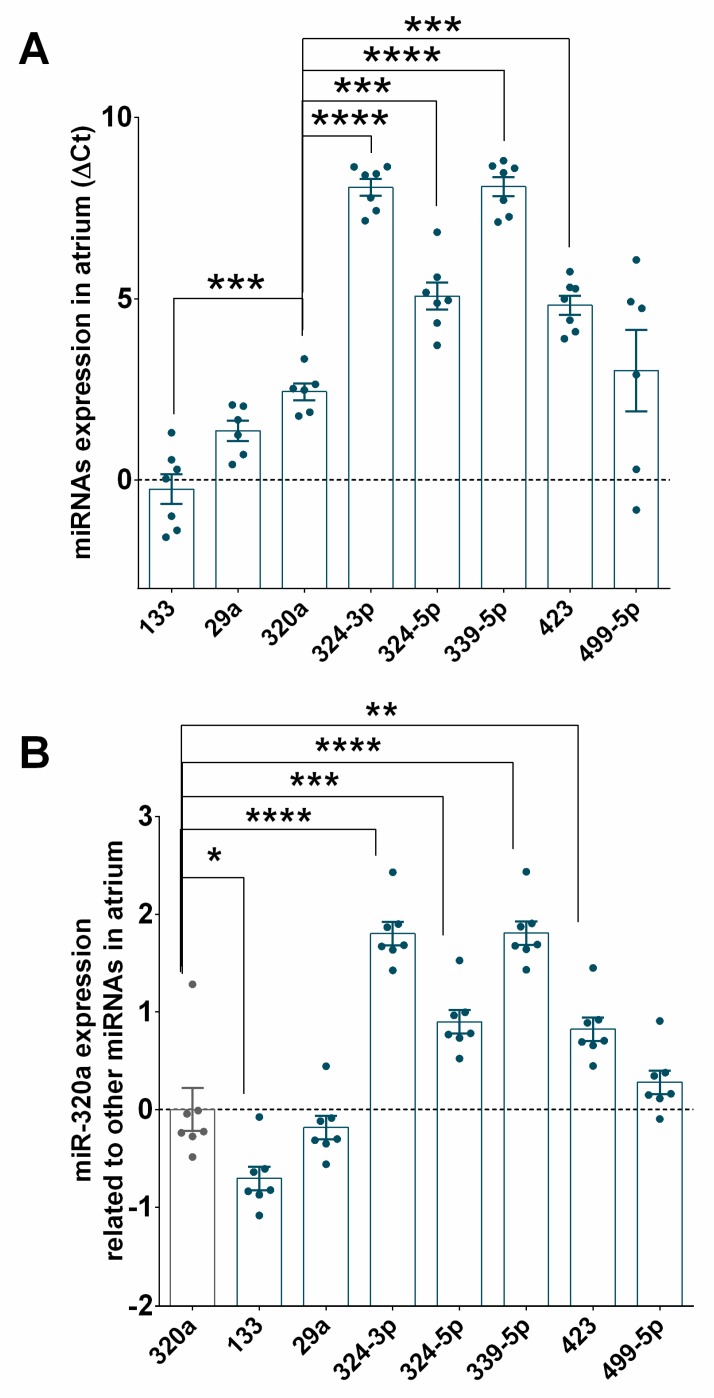
miR-320a is highly expressed in the atrium of patients with heart failure (HF) of ischaemic origin. (**A**) DeltaCt represents the level of Ct of different miRNAs compared to that of the endogenous control in the atrium of HF patients (*n* = 7). (**B**) Levels of expression of miR-320a related to the expression of the other miRNAs shown as fold change in logarithmic scale in the atrium of failing heart. Values are presented as the means ± SEM. Significance is indicated by (*) for *p* < 0.05, (**) for *p* < 0.01, (***) for *p* < 0.001), and (****) for *p* < 0.0001. Ordinary one-way ANOVA with multiples comparisons using T-test without correction (Fisher’s LSD test) was performed.

**Figure 9 jcm-09-01051-f009:**
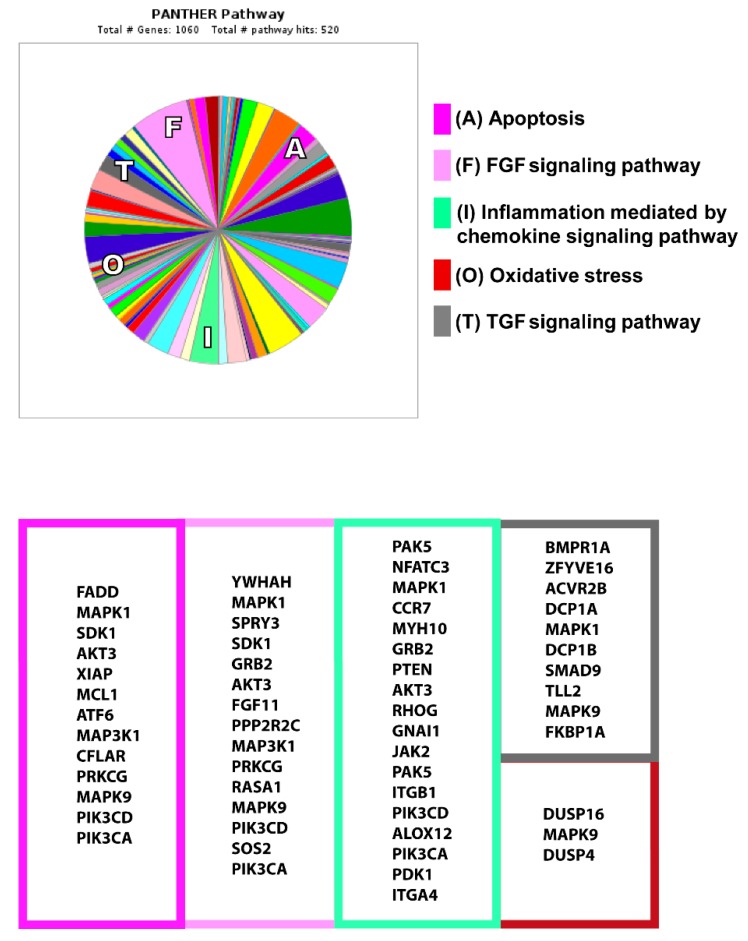
In silico analysis of miR-320a targets. PANTHER analysis showed the miR-320a predicted target genes involved in different pathways induced under ischaemia and reperfusion, such as apoptosis (dark pink), FGF (light pink) and TGF signalling (grey) pathways, inflammation (green) and oxidative stress (red).

**Table 1 jcm-09-01051-t001:** Demographic and clinical characteristics of STEMI patients (*n* = 42) and controls (*n* = 14). Data are shown as the means ± SEM and as n (%). Student’s t-tests were performed. LVEDV stands for the left ventricular end-diastolic volume; (*) indicates *p* < 0.05, which is considered statistically significant; ^$^ indicates that the variable was examined at admission.

	Controls (*n* = 14)	Patients (*n* = 42)	*P* Value
Age (years)	62.1 ± 11.6	58.4 ± 11.3	0.287
Male sex	7 (50.0%)	37 (88.1%)	**0.002 ***
Arterial hypertension	7 (50.0%)	21 (50.0%)	1.000
Smoking	7 (50.0%)	17 (40.5%)	0.541
Dyslipidaemia	8 (57.1%)	19 (45.2%)	0.449
Type 2 diabetes mellitus (%)	4 (28.6%)	10 (23.8%)	0.727
LVEDV (ml/m^2^) ^$^	50 ± 10.9	58.1 ± 12.7	0.162
Creatine kinase (mg/dL) ^$^	96.4 ± 60.1	2502 ± 2290.6	**>0.001 ***
Troponin T (ng/mL) ^$^	7.3 ± 4.9	6097.6 ± 5526.9	**0.003 ***

**Table 2 jcm-09-01051-t002:** Demographic and clinical characteristics of STEMI patients (*n* = 39). Patients were classified into two groups: no remodelling (*n* = 25) and remodelling (*n* = 14). Data are shown as the means ± SEM and as n (%). Student’s t-tests were performed. LVEDV stands for the left ventricular end-diastolic volume; (*) indicates that *p* < 0.05 is statistically significant; ^$^ indicates that the variable was examined at admission, ^$$^ at 1 month after PPCI and ^$$$^ at 6 months after PPCI.

	STEMI Patients (*n* = 39)	No remodelling (*n* = 25)	Remodelling (*n* = 14)	*P* Value
Age (years)	57.8 ± 10.4	57.2 ± 10.6	59.1 ± 10.3	0.596
Male sex	35 (89.7%)	23 (92.0%)	12 (85.7%)	0.547
Arterial hypertension	18 (46.2%)	10 (40.0%)	8 (57.1%)	0.316
Smoking	16 (41.0%)	11 (44.0%)	5 (35.7%)	0.625
Dyslipidaemia	17 (45.2%)	12 (48.0%)	5 (35.7%)	0.471
Type 2 diabetes mellitus	9 (23.1%)	6 (24.0%)	3 (21.4%)	0.860
LVEDV (ml/m^2^) ^$^	57.9 ± 12.8	58.9 ± 12.4	55.9 ± 13.6	0.487
LVEDV (ml/m^2^) ^$$$^	67.8 ± 26.0	56.9 ± 11.2	88.8 ± 33.4	**>0.001 ***
Creatine kinase (mg/dL) ^$^	3213.8 ±2209.0	2447.8 ± 1795.6	4526.9 ± 2289.9	**0.004 ***
Troponin T (ng/mL) ^$^	5806.6 ± 5404.1	4853.3 ± 4414.5	7508.9 ± 6672.3	0.143
Pro-BNP (pg/mL) ^$$^	1172.9 ± 710.7	1029.2 ± 896.2	1388.5 ± 270.4	0.466

**Table 3 jcm-09-01051-t003:** Multivariate logistic regression to determinate independent predictors of LVAR. OR: odds ratio; CI: confidence interval. (*) indicates *p* < 0.05, which is considered statistically significant; ^&^ miR-320a expression in patients’ serum at 1 month post-PPCI.

Factor	OR	95% CI	*P* Value
miR-320a ^&^	0.005	0.000–0.892	**0.045 ***
Age	1.075	0.929–1.243	0.330
Sex	<0.001	-	0.999
Creatine Kinase	1.001	1.000–1.002	0.189

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
