# Peer review of "Circulating miR-320a as a Predictive Biomarker for Left Ventricular Remodelling in STEMI Patients Undergoing Primary Percutaneous Coronary Intervention"

_jcm, 2020, doi:10.3390/jcm9041051_

Round 1
Reviewer 1 Report
Thank you for the opportunity to review the work of Galeano-Otero et al. They investigate the relation ship between miRNA expression and remodeling after STEMI. Additionally to their clinical observative data, they present some basic experiments underscoring the merit of their data. The manuscript is well written, the data presented clearly and the topic certainly of interest. I do have, however a simple suggestion to further increase the value of the manuscript.
The basic premise of the manuscript ist that miR-320a is associated with remodelling after 6 months. This is an interesting finding, however, given the sample size and many potential co-founders, could be a chance finding. I would, therefore, suggest performing a univariable and some multivariable logistic regression. Use "LVAR" as binary dependent variable and miR-320a at the distinct time-points as covariate; further >>at least<< age- and sex-adjustment should be done. Then the reader could estimate if miR-320a is an independent predictor of LVAR (and then I would suggest to the authors to state that proudly in the Title). If, however, this is not the case, this is a major limitation and should be stated and discussed honestly.
Author Response
"Please see the attachment." The pdf is a merge of the revised manuscript with highlighted changes, supplementary data, and point-by-point response to reviewer 1 comments.
Reviewer 2 Report
Dr Galeano-Otero et al studied the role of circulating microRNAs (MiRNAs) and its association with left ventricular adverse remodelling (LVAR) after S-T segment-elevation myocardial infarction (STEMI). They showed that miR-320a 39 was positively associated with LVAR . The topic is really interesting, the paper is well written, the graphs clear and esplicative.
Nonetheless I have two main concerns:
First of all is the small number in the remodelling group. I strongly doubt that it allows to reach a sufficient statistical power. From my quick calculations the assumptions are not so robust to support results.
Second, the analysis is simply a one-to-one comparison I believe that a more complete analysis with a higher number of patients would result in a very significant manuscript.
Would recommend authors collect more data and , with the help of a statistician rework the manuscript
Author Response
"Please see the attachment." The pdf is a merge of the revised manuscript with highlights changes, supplementary data, and point-by-point response to reviewer 2 comments.

Round 2
Reviewer 1 Report
Happy with the changes!
Author Response
We wish to thank our valuable reviewer for his/her comments. We are glad that he is happy with the improvement of this manuscript.
Reviewer 2 Report
The authors tried to address the points raised by me and the other review.
I am still convinced that the small number of patients is the main drawback of the paper.
Nonetheless, I have greatly appreciated the effort by the authors
Author Response
First of all, we would like to thank the Reviewer for his/her constructive comments and for recognizing our efforts in revising this manuscript. As stated in my previous reply, we are still working in this research line and taking in consideration our promising data confirming the results presented in this manuscript, we hope we will provide new data in larger number of patients in very close future.